# *Soft Thinking*: Unlocking the Reasoning Potential of LLMs in Continuous Concept Space

**Zhen Zhang[1]\* Xuehai He[2]\* Weixiang Yan[1]**
**Ao Shen[4] Chenyang Zhao[3,5] Xin Eric Wang[1]**
[1]University of California, Santa Barbara, [2]University of California, Santa Cruz
[3]University of California, Los Angeles, [4]Purdue University, [5]LMSYS Org
zhen_zhang@ucsb.edu, ericxwang@ucsb.edu

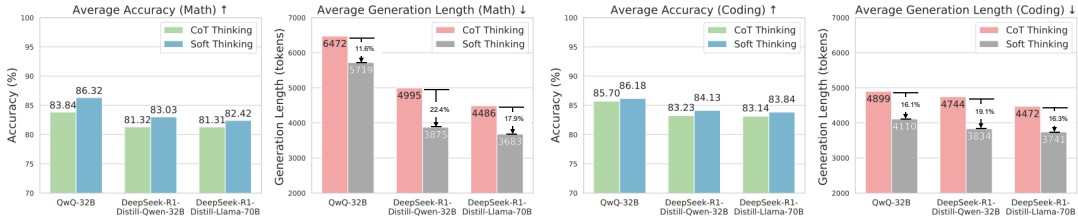

Figure 1: *Soft Thinking* vs. Chain-of-Thought thinking on mathematical and coding datasets. *Soft Thinking* consistently improves both accuracy (with improvements of up to **2.48%** on pass@1 accuracy) and generation efficiency (achieving up to **22.4%** reduction in generation length) across both tasks, **without any training**.

## Abstract

Human cognition typically involves thinking through abstract, fluid concepts rather than strictly using discrete linguistic tokens. Current reasoning models, however, are constrained to reasoning within the boundaries of human language, processing discrete token embeddings that represent fixed points in the semantic space. This discrete constraint restricts the expressive power and upper potential of such reasoning models, often causing incomplete exploration of reasoning paths, as standard Chain-of-Thought (CoT) methods rely on sampling one token per step. In this work, we introduce *Soft Thinking*, a training-free method that emulates human-like "soft" reasoning by generating soft, abstract *concept tokens* in a continuous concept space. These *concept tokens* are created by the probability-weighted mixture of token embeddings, which form the continuous concept space, enabling smooth transitions and richer representations that transcend traditional discrete boundaries. In essence, each generated *concept token* encapsulates multiple meanings from related discrete tokens, implicitly exploring various reasoning paths to converge effectively toward the correct answer. Empirical evaluations on diverse mathematical and coding benchmarks consistently demonstrate the effectiveness and efficiency of *Soft Thinking*, improving pass@1 accuracy by up to 2.48 points while simultaneously reducing token usage by up to 22.4% compared to standard CoT. Qualitative analysis further reveals that *Soft Thinking* outputs remain highly interpretable and readable, highlighting the potential of *Soft Thinking* to break the inherent bottleneck of discrete language-based reasoning. Code is available here.

---

\*Equal contribution

39th Conference on Neural Information Processing Systems (NeurIPS 2025).

*The limits of my language mean the limits of my world.*

—Ludwig Wittgenstein

# 1   Introduction

Large Language Models (LLMs) have achieved impressive results across a wide range of complex reasoning tasks. A key technique contributing to this success is Chain-of-Thought (CoT) reasoning [1–4], which enables models to solve problems step-by-step by generating intermediate reasoning steps in natural language. Despite its effectiveness, standard CoT reasoning restricts the model's outputs to sequences of discrete, predefined tokens, inherently bounded by the expressive limits of human language. Confining LLM reasoning to the discrete natural language tokens may fundamentally restrict their potential to represent and manipulate abstract concepts. Moreover, neuroscientific evidence shows that the human brain represents and stores information at the level of abstract concepts, not merely words, and that reasoning is based on non-verbal conceptual processing independent of the language network [5–8].

Another fundamental limitation of standard CoT reasoning is its inherently unidirectional and sequential nature: at each step, the model samples a single token, committing to one specific branch of the reasoning path. In tasks with high uncertainty or multiple plausible trajectories, this approach can easily lead the model down an incorrect path, resulting in suboptimal answers or wasted tokens on the wrong path, thus reducing both performance and token efficiency [9, 10]. In contrast, humans do not rely solely on sequentially producing explicit linguistic tokens. Instead, they can simultaneously consider multiple possibilities, integrate abstract concepts, and only later verbalize their thoughts. This allows for more flexible, parallel, and comprehensive reasoning, enabling humans to navigate complex problems more effectively.

In this work, we propose a new perspective: instead of constraining LLMs to reason within the discrete, sequential space of language tokens, we aim to enable LLMs to reason with soft, abstract concepts, which encompass more general and fine-grained semantics and retain information about multiple possible paths. To achieve this, we introduce *Soft Thinking*, a training-free method that unlocks the reasoning potential of LLMs in a continuous concept space. Specifically, *Soft Thinking* replaces the discrete token selection in standard CoT with probabilistic soft aggregation over the entire vocabulary, which we refer to as a *concept token*. This retains the original distribution of the next step. At each step, we construct a new embedding from a concept token by probability-weighting all token embeddings, which form the continuous concept token. This approach allows the model to represent and process abstract concepts, endowing each output token with more nuanced and fine-grained semantics, and enabling the processing of multiple paths conceptually.

Unlike standard CoT that forces the model to commit to a single next token at each step by collapsing the probability distribution, our method naturally preserves a "superposition" which retains the entire information in each step. As a result, we introduce a *Cold Stop* mechanism to further boost efficiency and address the challenge of generation collapse (e.g., repetition) caused by out-of-distribution (OOD) [11] inputs, where certain *concept tokens* may be unseen during training. To be specific, *Cold Stop* monitors the entropy of the model's output distribution at each step and terminates the reasoning process early when the model demonstrates high confidence (i.e., low entropy) over several consecutive steps. This mechanism prevents unnecessary computation and mitigates the risk of model collapse when dealing with OOD inputs, ensuring more robust and efficient reasoning.

*Soft Thinking* offers two major advances. First, by operating in the continuous concept space formed as a convex combination of all token embeddings, the model can capture and manipulate abstract concepts and detailed semantic information; Second, because each concept token keeps a probability distribution from all possible next tokens, the model can implicitly and efficiently explore multiple reasoning paths in parallel, rather than being limited to a single trajectory. Therefore, *Soft Thinking* not only improves the comprehensiveness of reasoning but also accelerates convergence toward correct answers.

Empirical evaluations conducted on mathematical and coding benchmarks using mainstream LLM architectures, including Llama [12] and Qwen [13] with large model sizes 32B and 70B parameters, consistently demonstrate the effectiveness and efficiency of *Soft Thinking*. The method improves

pass@1 accuracy by up to 2.48 points while simultaneously reducing token usage by up to 22.4% compared to standard CoT. Furthermore, qualitative assessments reveal that intermediate reasoning steps generated by *Soft Thinking* are highly readable, interpretable, and informative. Overall, *Soft Thinking* presents an alternative reasoning paradigm that breaks the bottleneck of discrete token-based reasoning.

## 2 Related Work

**Chain-of-Thought (CoT) Reasoning.**   CoT reasoning enhances the multi-step reasoning capabilities of large language models by introducing explicit intermediate steps. Existing approaches primarily include prompt-based learning methods [1–3], supervised fine-tuning [14, 15], and reinforcement learning optimization [16–19]. Moreover, according to inference-time scaling laws [20], model performance continues to improve as the length of reasoning chains increases. However, as the chain grows longer, the computational cost also rises, making efficiency a growing concern. To address this challenge, we propose to shift CoT reasoning from the discrete natural language token to a continuous concept space, which is formed as a convex combination of all token embeddings. In this space, the model can select and integrate multiple potential reasoning trajectories at the token level.

**Continuous Space Reasoning.**    [21] constructed datasets for two-hop reasoning tasks and showed that intermediate reasoning variables could be decoded from hidden representations. Building on this, [22] introduced interventions on hidden states to manipulate reasoning outcomes. Parallel latent reasoning paths have also been observed [23]. [24] introduced latent planning steps: proposed predicting discrete planning tokens before generating reasoning steps. [25] proposes to do reasoning at an abstract language level beyond tokens and explores an explicit hierarchical structure. [26] proposes extracting text embeddings from the last token of LLMs fine-tuned with instructions on contrastive data. COCONUT [27] uses the last hidden state of the model's final layer as the next-step embedding. However, this method still face critical challenges. In language models with fewer than 7B parameters, the input embedding layer and the output language model head are typically weight-tied, enabling continuous-space reasoning by aligning the input and output spaces after extensive training. In contrast, for models with more than 7 billion parameters, these components are typically decoupled, meaning that the hidden states and input embeddings reside in different spaces. Directly using hidden states as input embeddings leads to significant representational mismatch, which is difficult to bridge even with extensive retraining. Such retraining often leads to overfitting, catastrophic forgetting, or ineffective performance in practice [28]. To address these limitations, we propose a training-free approach that utilizes the distribution over the vocabulary at each step as a bridge. This method effectively aligns the hidden state output space with the input embedding space, enabling seamless representation alignment during continuous-space reasoning.

## 3 Methodology

In this section, we introduce *Soft Thinking*, a method that generalizes standard Chain-of-Thought (CoT) reasoning by replacing discrete one-hot tokens with *concept tokens* and keeping the entire original probability distribution. As shown in Figure 2, the new embeddings are computed using probability-weighted interpolation across all embeddings based on the preceding *concept token*, facilitating reasoning within a continuous concept space. Furthermore, we propose the *Cold Stop* mechanism, which halts intermediate reasoning steps when overconfident, enhancing inference efficiency and preventing generation collapse.

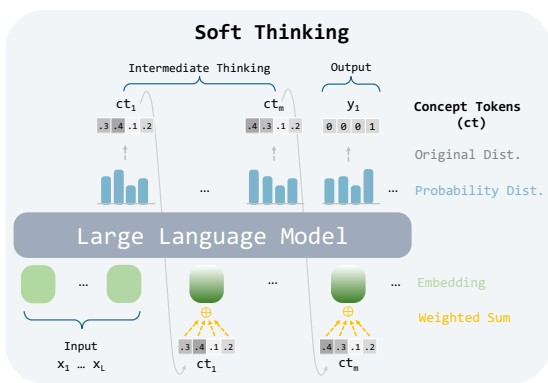

Figure 2: *Soft Thinking* replaces discrete tokens with soft, abstract *concept tokens*, enabling reasoning in continuous concept space.

## 3.1 Preliminary: Standard Chain-of-Thought Decoding

Let $V$ be the vocabulary of size $|V|$, and let $E \in \mathbb{R}^{|V| \times d}$ be the token embedding matrix. For any token index $k$, we denote its embedding by $e(k) = E[k] \in \mathbb{R}^d$. Given an input context: $x_{1:L} = (x_1, x_2, \ldots, x_L)$, the model first generates an intermediate reasoning trace (the Chain-of-Thought) of length $m$: $t_{1:m} = (t_1, t_2, \ldots, t_m)$, and then produces a final answer of length $n$: $y_{1:n} = (y_1, y_2, \ldots, y_n)$.

At each intermediate thinking step $i$, the LLM consumes the embeddings of all previously generated tokens and the input, and the next token is then sampled discretely:

$$t_i \sim p_i = \text{LLM}\big(e(x_{1:l}),\ e(t_{1:i-1})\big) \in \Delta^{|V|-1}, \tag{1}$$

Here, $\Delta^{|V|-1}$ denotes the $(|V|-1)$-dimensional probability simplex, representing the set of all valid probability distributions over the vocabulary. Decoding continues until the special end-of-thinking token $\langle/\text{think}\rangle$ is generated, i.e. $t_m = \text{encode}(\langle/\text{think}\rangle)$.

After reasoning, the model switches to answer mode. For each answer position $j$, it computes the probability $q_j$ and then sample one token $y_j$:

$$y_j \sim q_j = \text{LLM}\big(e(x_{1:l}),\ e(t_{1:m}),\ e(y_{1:j-1})\big) \in \Delta^{|V|-1}, \tag{2}$$

All tokens in both stages are drawn from their respective discrete distributions, committing to a discrete token id at each step. In the next section, we introduce *Soft Thinking*, which replaces this discrete sampling with continuous *concept tokens*, preserving the full distributional information throughout multi-step reasoning.

## 3.2 *Soft Thinking*: Reasoning in a Continuous Concept Space

**Definition 1** (*Concept Token*). At any intermediate thinking step, let $p \in \Delta^{|V|-1}$ be the LLM-produced probability distribution over the vocabulary. We call this probability vector a *concept token*, denoted by

$$ct := p. \tag{3}$$

Unlike a traditional step that collapses the distribution to a single token id, the concept token preserves the full distribution of every possible next step.

**Definition 2** (*Continuous Concept Space*). Let $E \in \mathbb{R}^{|V| \times d}$ be the embedding matrix and $e(k) = E[k]$ the embedding of the $k$-th vocabulary item. The *continuous concept space* is the convex combination of all embedding vectors.

$$\mathcal{C} = \left\{ \sum_{k=1}^{|V|} \alpha_k\, e(k)\ :\ \alpha \in \Delta^{|V|-1} \right\} \subset \mathbb{R}^d, \tag{4}$$

i.e. the set of all probability-weighted mixtures of token embeddings. Note that this is different from the usual semantic space, which is modeled as a $d$-dimensional real vector space.

**Reasoning Process.** *Soft Thinking* only replaces the intermediate thinking step of the standard CoT. At each step of soft thinking, the model generates a *concept token* defined by Definition 1. Then, in the next step, the *concept token* $ct$ is injected back into the LLM by the embedding of *concept token*:

$$\tilde{e}_{\text{next}} = \sum_{k=1}^{|V|} ct[k]\, e(k) = \sum_{k=1}^{|V|} p[k]\, e(k) \in \mathcal{C}. \tag{5}$$

When the most probable token for a certain concept token is the end-of-thinking, the intermediate reasoning process stops, and the model switches to generating the output. All output stage tokens $y_j$ are sampled in the usual discrete manner; only the intermediate thinking phase flows through the continuous concept space defined above.

**Why *Soft Thinking* Helps.** Using *concept tokens* allows the model to avoid making hard decisions too early. Instead of selecting a single token at each step, the model keeps the full probability distribution over vocabulary. This gives it the flexibility to explore different reasoning paths, especially when it's unsure. By working in this *continuous concept space*, the model can represent more abstract concepts that don't map cleanly to a single word. These abstract concepts can later evolve into more concrete thoughts as reasoning continues (see Figure 4). This flexibility helps the model think more clearly, avoid early mistakes, and better handle complex multi-step problems.

***Cold Stop*** While *concept tokens* enable more abstract reasoning, feeding in continuous *concept tokens* during inference places the model in an out-of-distribution (OOD) regime. This can lead to model collapse if the reasoning process continues for too long without correction. To mitigate this, we propose a *Cold Stop* mechanism that dynamically stops intermediate reasoning when the model becomes overconfident. At each step, we compute the entropy of the *concept token* :

$$H(p) = -\sum_{k=1}^{|V|} p[k] \log p[k].$$ (6)

Since *Soft Thinking* preserves the entire probability distribution at each step, the entropy serves as a natural signal for uncertainty, which is often used in LLMs to evaluate the quality of generation [29]. Low entropy, typically represents "cold" in physics, indicates that the model is confident in its prediction [30], and thus can conclude soon. Given an entropy threshold $\tau$ and a required number of consecutive confident steps $k$, we apply the following rule:

- If $H(p) < \tau$, increment a low-entropy step counter; otherwise, reset the counter.
- When the counter reaches $k$, we insert an end-of-thinking token $\langle /\text{think} \rangle$ to conclude reasoning and begin final answer generation.

This strategy avoids unnecessary computation and prevents the model collapse under OOD conditions, while preserving the benefits of soft thinking through an entropy-based confidence measure.

**Complexity Analysis.** *Soft Thinking* incurs only two lightweight additions to standard CoT. When computing each concept token embedding, we first apply a top-$k$ top-$p$ filter to the token distribution to remove low-probability noise, select top-$n$ tokens with highest probability, renormalize, and then perform a single dense matrix–vector multiplication over the filtered subset, resulting in $O(n \cdot d)$ computational cost per reasoning step (where $d$ the embedding dimension). Calculating the entropy for *Cold Stop* requires $O(|V|)$ time, but adds negligible overhead compared to a model forward pass.

### 3.3 Theoretical Analysis

In this section, we provide a theoretical analysis showing how *Soft Thinking* approximates the full path-summation of standard Chain-of-Thought (CoT) by iteratively constructing linear surrogate representations via *concept tokens*. We begin by rewriting the exact expansion of the marginal likelihood and then derive a sequence of linearization steps that culminate in the continuous *concept token* approximation.

**Exact Path-Summation.** Let $x = x_{1:l}$ denote the input context and $t_{1:m}$ be the first $m$ intermediate reasoning tokens. The true probability of a final answer $y = y_{1:n}$ is obtained by marginalizing over all possible reasoning trajectories of length $m$:

$$p(y \mid x) = \sum_{t_1} p(t_1 \mid x) \left( \sum_{t_2} p(t_2 \mid x, t_1) \cdots \left( \sum_{t_m} p(t_m \mid x, t_{1:m-1}) \, p(y \mid x, t_{1:m}) \right) \right). \quad (7)$$

This expansion entails an exponential number of paths, since each summation runs over the full vocabulary $V$.

**First-Order Linearization.** Focusing on the outermost summation,

$$p(y \mid x) = \sum_{t_1} p(t_1 \mid x) p(y \mid x, t_1). \quad (8)$$

Viewing the sampled token $t_1$ as an one-hot vector $t_1 \in \{0,1\}^{|V|}$ with $\sum_k t_{1,k} = 1$, its expectation under the multinomial distribution is the first *concept token*:

$$ct_1 = \mathbb{E}[t_1] = \sum_{t_1} p(t_1 \mid x) \, t_1 = p(\cdot \mid x) \in \Delta^{|V|-1}, \quad (9)$$

which aligns the *concept token* in Definition 1.

By a linear approximation of $p(y \mid x, \cdot)$ around the mean, we obtain

$$p(y \mid x) = \sum_{t_1} p(t_1 \mid x) \, p(y \mid x, t_1) \approx p\big(y \mid x, \sum_{t_1} p(t_1 \mid x) \, t_1\big) = p\big(y \mid x, ct_1\big). \qquad (10)$$

Thus, the full outer summation is replaced by a single evaluation at the *concept token* $ct_1$.

**Recursive Approximation.** We now apply the same linearization recursively. Given $ct_1$, the conditional probability expands as

$$p(y \mid x, ct_1) = \sum_{t_2} p(t_2 \mid x, ct_1) \, p\big(y \mid x, ct_1, t_2\big) \approx p\big(y \mid x, ct_1, ct_2\big), \qquad (11)$$

where $ct_2 = \sum_{t_2} p(t_2 \mid x, ct_1) \, t_2$.

Repeating this process for all $m$ steps yields the continuous expansion:

$$p(y \mid x) \approx p\big(y \mid x, ct_1, ct_2, \ldots, ct_m\big). \qquad (12)$$

**Comparison to Standard CoT.** In contrast, discrete CoT replaces each summation $\sum_{t_j} p(t_j \mid \cdot)$ with sampling a single token, thereby discarding mass from all other paths. *Soft Thinking* preserves the full probability distribution at each step through *concept tokens*, collapsing the exponential path-summation in Eq. 7 into a single forward pass under a sequence of linear approximations.

## 4 Experiments & Results

### 4.1 Experiment Setup

**Benchmarks.** We conduct a comprehensive evaluation of our method on eight benchmark tasks, including Math500 [31], AIME 2024 [32], GSM8K [33], and GPQA-Diamond [34] in the mathematics domain, as well as HumanEval [35], MBPPMBPP [36], and LiveCodeBench [37] in the programming domain. Detailed descriptions of these benchmarks are provided in Appendix A.2.

**Models.** We select three widely used open-source LLMs: QwQ-32B [13], DeepSeek-R1-Distill-Qwen-32B [38], and DeepSeek-R1-Distill-Llama-70B [38]. This diverse selection is designed to demonstrate the effectiveness and generalizability of the *Soft Thinking* approach across different model scales (32B and 70B), model architectures (Qwen and LLaMA), and training paradigms (QwQ-32B is trained with reinforcement learning, while the DeepSeek models are trained via supervised distillation).

**Baseline Methods.** We evaluate the performance of *Soft Thinking* by comparing it with two representative baselines. These baselines include Standard CoT Thinking, which employs explicit step-by-step reasoning, and Standard Greedy CoT Thinking, which utilizes greedy decoding at each step of the reasoning process.

**Metrics.** We use the Pass@1 metric to evaluate the accuracy of the model's generated answers. The formula for computing Pass@k is as follows:

$$\text{Pass@k} = 1 - \frac{\binom{n-c}{k}}{\binom{n}{k}} \qquad (13)$$

where $n$ is the total number of samples (e.g., 16), $c$ is the number of correct samples, and $k$ is the number of samples selected (we set $k = 1$). Therefore, $\text{Pass@1} = \frac{c}{n}$. Besides, we evaluate the model's reasoning efficiency by reporting the number of tokens generated specifically for correct solutions. These two metrics allow us to comprehensively evaluate the trade-off between computational cost and performance across different methods.

**Implementation Details.** We reuse the model's existing embedding matrix without any extra parameters or layers as *concept tokens*, and the *Cold Stop* controller monitors decoder entropy and emits an end-of-thinking marker when triggered. No model weights update, architecture change, or additional training procedures, *Soft Thinking* can be plugged into the CoT pipeline of any LLM with minimal engineering effort. We implement our *Soft Thinking* on SGLang [39], enabling fast inference (see Appendix A.3 for implementation details). We evaluate our method on a server equipped with eight NVIDIA H100 80GB GPUs.

| | Accuracy ↑ | | | | | Generation Length ↓ | | | | |
|---|---|---|---|---|---|---|---|---|---|---|
| | MATH 500 | AIME 2024 | GSM8K | GPQA Diamond | Avg. | MATH 500 | AIME 2024 | GSM8K | GPQA Diamond | Avg. |
| QwQ-32B [13] | | | | | | | | | | |
| CoT Thinking | 97.66 | 76.88 | 96.67 | 64.17 | 83.84 | 4156 | 12080 | 1556 | 8095 | 6472 |
| CoT Thinking (Greedy) | 97.00 | 80.00 | 96.57 | 65.15 | 84.68 (↑ 0.84) | 3827 | 11086 | 1536 | 7417 | 5967 (↓ 7.8%) |
| Soft Thinking | **98.00** | **83.33** | **96.81** | **67.17** | **86.32** (↑ 2.48) | **3644** | **10627** | **1391** | **7213** | **5719** (↓ 11.6%) |
| DeepSeek-R1-Distill-Qwen-32B [38] | | | | | | | | | | |
| CoT Thinking | 94.50 | 72.08 | 95.61 | 63.10 | 81.32 | 3543 | 9347 | 875 | 6218 | 4995 |
| CoT Thinking (Greedy) | 93.00 | 63.33 | 95.30 | 59.09 | 77.68 (↓ 3.64) | 3651 | 8050 | 1048 | 8395 | 5286 (↑ 5.8%) |
| Soft Thinking | **95.00** | **76.66** | **95.83** | **64.64** | **83.03** (↑ 1.71) | **3373** | **6620** | **785** | **4722** | **3875** (↓ 22.4%) |
| DeepSeek-R1-Distill-Llama-70B [38] | | | | | | | | | | |
| CoT Thinking | 94.70 | 70.40 | 94.82 | 65.34 | 81.31 | 3141 | 8684 | 620 | 5500 | 4486 |
| CoT Thinking (Greedy) | 94.61 | **73.33** | 93.60 | 66.16 | 81.92 (↑ 0.61) | **2877** | 9457 | 606 | **4443** | 4345 (↓ 3.1%) |
| Soft Thinking | **94.80** | **73.33** | **94.90** | **66.66** | **82.42** (↑ 1.11) | 3021 | **6644** | **597** | 4470 | **3683** (↓ 17.9%) |

Table 1: Comparison of *Soft Thinking* and various baseline methods on accuracy and generation length across mathematical datasets. Best results are highlighted in **bold**.

| | Accuracy ↑ | | | | Generation Length ↓ | | | |
|---|---|---|---|---|---|---|---|---|
| | HumanEval | MBPP | LiveCodeBench | Avg. | HumanEval | MBPP | LiveCodeBench | Avg. |
| QwQ-32B [13] | | | | | | | | |
| CoT Thinking | 97.63 | 97.49 | 62.00 | 85.70 | 2557 | 2154 | 9986 | 4899 |
| CoT Thinking (Greedy) | 95.73 | 96.50 | 57.35 | 83.19 (↓ 2.51) | **2396** | **2069** | **7034** | **3833** (↓ 21.8%) |
| Soft Thinking | **98.17** | **97.66** | **62.72** | **86.18** (↑ 0.48) | 2638 | 2157 | 7535 | 4110 (↓ 16.1%) |
| DeepSeek-R1-Distill-Qwen-32B [38] | | | | | | | | |
| CoT Thinking | 97.25 | 95.13 | 57.33 | 83.23 | 3095 | 2761 | 8376 | 4744 |
| CoT Thinking (Greedy) | 87.19 | 87.54 | 43.36 | 72.70 (↓ 10.53) | **2294** | **1703** | **4702** | **2900** (↓ 38.9%) |
| Soft Thinking | **97.56** | **95.33** | **59.50** | **84.13** (↑ 0.90) | 2713 | 2534 | 6255 | 3834 (↓ 19.1%) |
| DeepSeek-R1-Distill-Llama-70B [38] | | | | | | | | |
| CoT Thinking | 97.71 | 94.77 | 56.94 | 83.14 | 2711 | 2386 | 8319 | 4472 |
| CoT Thinking (Greedy) | 92.07 | 91.82 | 48.02 | 77.30 (↓ 5.84) | 2192 | 1979 | 5438 | 3203 (↓ 28.3%) |
| Soft Thinking | **98.17** | **94.94** | **58.42** | **83.84** (↑ 0.70) | 2498 | 2214 | 6512 | 3741 (↓ 16.3%) |

Table 2: Comparison of *Soft Thinking* and various baseline methods on accuracy and generation length across two coding datasets. Best results are highlighted in **bold**.

## 4.2 Hyper-parameter Settings

For all experiments, the maximum generation length was set to $32,768$, the temperature to $0.6$, top-$k$ to $30$, and top-$p$ to $0.95$, unless specified otherwise. The Standard CoT baseline was evaluated using 16 samples per problem to calculate Pass@1 accuracy, whereas the greedy CoT approach utilized a temperature of 0 with a single sample.

For *Soft Thinking*, the *concept token* was determined using the top-$n$ tokens, where $n \in 5, 10, 15, 20, 30$, along with an entropy threshold $\tau$ chosen from $0.01, 0.05, 0.1, 0.2$ and a length threshold $k$ selected from $128, 256, 512, 1024$. All other settings were kept consistent. We find that $n = 15$ yields the best performance for QwQ-32B [13], while $n = 10$ is optimal for DeepSeek-R1 models [38]. Results are reported based on the best-performing combinations of $\tau$ and $k$.

## 4.3 Results and Analysis

We present the quantitative evaluation results of *Soft Thinking* and other baseline methods on mathematical and coding datasets in Table 1 and Table 2, respectively.

**Improved Pass@1 Accuracy.** Our proposed *Soft Thinking* consistently enhances Pass@1 accuracy across all evaluated math and coding benchmarks, demonstrating its broad effectiveness and generalization ability. For instance, on mathematical reasoning tasks, the QwQ-32B model's average Pass@1 improves from 83.84% (CoT Thinking) to 86.32% (*Soft Thinking*), representing a notable gain of 2.48% points. On the challenging AIME2024 dataset, the improvement reaches 6.45% points. Similarly, for DeepSeek-R1-Distill-Qwen-32B and DeepSeek-R1-Distill-Llama-70B, Pass@1 increases by 1.71% and 1.11% points, respectively. On coding benchmarks, *Soft Thinking* also achieves consistent improvements: QwQ-32B sees a 0.48-point increase in average Pass@1, while DeepSeek-R1-Distill-Qwen-32B and DeepSeek-R1-Distill-Llama-70B improve by 0.90% and 0.70%

Figure 3: A comparison between standard CoT and *Soft Thinking* on a multiplication problem. We select the token with the highest probability at each step of *Soft Thinking* for readability and interpretability. Full distribution is visualized in Figure 4. Red text denotes repetitive, useless words.

points, respectively. These results demonstrate that *Soft Thinking* provides robust accuracy gains across both math and code domains.

**Token Efficiency.** A key advantage of *Soft Thinking* is its significant reduction in generation length, leading to improved token efficiency. On mathematical reasoning benchmarks, *Soft Thinking* reduces token usage for QwQ-32B by 11.6%, DeepSeek-R1-Distill-Qwen-32B by 22.4%, and DeepSeek-R1-Distill-Llama-70B by 17.9% compared to standard CoT thinking. Similar trends are observed in coding tasks: for QwQ-32B, DeepSeek-R1-Distill-Qwen-32B, and DeepSeek-R1-Distill-Llama-70B, token usage is reduced by 16.1%, 19.1%, and 16.3%, respectively. This substantial reduction in token usage not only translates to lower computational and inference costs, but also indicates that the model can reach correct answers through more concise and efficient reasoning. Such token efficiency is particularly valuable for real-world applications, where cost, speed, and scalability are crucial.

**Analysis.** Our experimental results demonstrate that *Soft Thinking* achieves simultaneous improvements in both reasoning performance and token efficiency across a diverse set of mathematical and coding benchmarks. This dual gain highlights a key advantage of our approach: by leveraging *concept tokens* that encode richer semantic information at each reasoning step, the model is able to represent and process more abstract or composite ideas within a single token. As a result, fewer reasoning steps are required to reach the correct solution, directly translating to reduced token usage. Results also stand in stark contrast to the greedy decoding baseline, which, while also reducing token usage, suffers from substantial drops in accuracy, particularly on complex code generation tasks where the loss of diversity in reasoning paths leads to premature convergence on suboptimal solutions.

These findings suggest that the superior efficiency of *Soft Thinking* is not simply a result of more aggressive pruning or shortcutting, but rather reflects a fundamental enhancement in the model's reasoning process. By maintaining the full probability distribution over possible next tokens at each step, our method allows for a "soft" aggregation of multiple reasoning trajectories, effectively broadening the model's exploration space without incurring the combinatorial explosion of explicit enumeration. This enables the model to make more informed and confident decisions earlier in the reasoning chain, reducing unnecessary detours and redundant steps.

Overall, the results provide strong evidence that *Soft Thinking* breaks the traditional trade-off between performance and efficiency in large language model reasoning. Instead of sacrificing one for the other, our approach inherently boosts both, offering a more powerful and concise reasoning framework that is readily applicable to a wide range of tasks and model architectures.

### 4.4 Qualitative Results

**Visualization of Shortened Examples.** Figure 3 demonstrates the comparison between Standard CoT and *Soft Thinking*. We select the token with the highest probability at each step of *Soft Thinking* for visualization. It can be seen that *Soft Thinking* has high readability and interpretability. While both methods arrive at the correct answer $(1, 462)$, Soft Thinking produces a significantly more concise explanation (96 tokens vs. 157 tokens). This demonstrates Soft Thinking's ability to preserve logical structure while improving token efficiency.

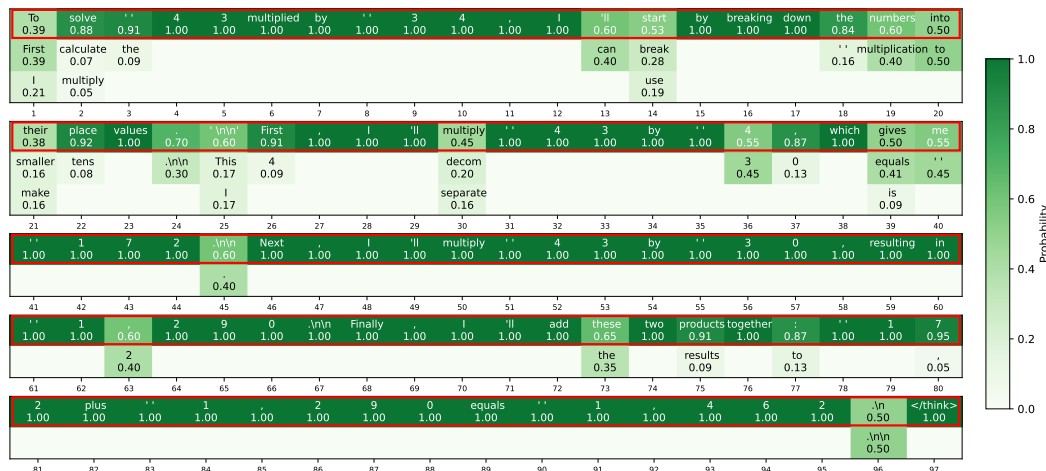

Figure 4: An example illustrating the probability distribution of our proposed *Soft Thinking* method. At each step, top-$k$ token candidates and their probabilities are shown. Red boxes indicate the selected tokens that form the final generated sequence for readability and interpretability.

| | Accuracy | | Generation Length All | | Generation Length Correct | |
|---|---|---|---|---|---|---|
| | AIME 2024 ↑ | LiveCodeBench ↑ | AIME 2024 ↓ | LiveCodeBench ↓ | AIME 2024 ↓ | LiveCodeBench ↓ |
| COCONUT-TF | 0.0 | 0.0 | 32,768 | 32,768 | – | – |
| Average Embedding | 6.66 | 7.49 | 30,802 | 30,474 | **6,556** | **2,141** |
| Soft Thinking w/o Cold Stop | 73.33 | 56.98 | 12,991 | 13,705 | 9,457 | 6,877 |
| Soft Thinking w/ Cold Stop | **83.33** | **62.72** | **11,445** | **12,537** | 10,627 | 7,535 |

Table 3: Ablation study of Soft Thinking with QwQ-32B on the AIME 2024 and LiveCodeBench. COCONUT-TF represents training-free COCONUT [27].

**Visualization of Embedding Weights.** In Figure 4, we present the token probability distributions at each intermediate reasoning step of *Soft Thinking*. For demonstration, we highlight the top three tokens. During exploratory reasoning phases, such as steps $1-3$, $13-14$, and $18-20$, the token distribution appears more uniform, reflecting the presence of multiple viable paths. In contrast, during precise calculations, the token distribution becomes nearly one-hot, indicating that textual elements facilitate path exploration, while numerical components handle exact computations. Notably, at steps 36-37, the model evaluates whether to multiply by 4 or 30, ultimately assigning a higher probability to 4. As a result, at step 42, the model selects multiplication by 4. This demonstrates how Soft Thinking integrates path exploration across consecutive *concept tokens*, thereby enhancing both reasoning flexibility and depth.

## 4.5 Ablation Study

To comprehensively assess the effectiveness of *Soft Thinking*, we conducted ablation studies focusing on different strategies for *concept token* and the impact of the *Cold Stop* mechanism.

**Different strategies for *concept token*.** Specifically, we compared (1) the training-free COCONUT approach [27], which directly feeds the previous hidden state as the next input embedding; (2) a simple average embedding strategy that takes the mean of the top-$n$ token embeddings (we use 5 for ablation); and (3) our *Soft Thinking*, which computes a probability-weighted over embeddings. As shown in Table 3, we observe that the training-free COCONUT fails entirely, producing no correct answers and always reaching the maximum generation length. The average embedding method performs marginally better, yielding a small number of correct solutions but still suffering from extremely long output. In contrast, our *Soft Thinking* substantially improves both accuracy and token efficiency.

**Impact of *Cold Stop*.** We further analyze the *Cold Stop* by comparing *Soft Thinking* with and without it. Without *Cold Stop*, the model is prone to generation collapse, especially due to out-of-

distribution (OOD) issues because models have never been trained on *concept tokens*. It will begin to repeat until it hits the maximum generation length. This OOD-induced collapse significantly increases the average generation length across all problems. However, for problems that are solved correctly (typically those requiring shorter reasoning chains), the average length remains relatively low, as these cases are less likely to trigger collapse. When *Cold Stop* is activated, generation collapse is effectively mitigated, and the model avoids unnecessary exploration along overconfident paths, resulting in a significant reduction in average generation length for all problems. Interestingly, as *Cold Stop* allows the model to correctly solve more challenging problems that require longer reasoning chains, the average length for correct solutions increase. Nevertheless, as demonstrated in Tables 1 and 2, *Soft Thinking* with *Cold Stop* not only solves more problems than standard CoT but also achieves greater overall efficiency, confirming the effectiveness of our approach.

## 5  Conclusion

In this work, we present *Soft Thinking*, a novel, training-free framework that enables large language models to reason in a continuous concept space by leveraging probabilistically weighted *concept tokens* instead of traditional discrete tokens. By aggregating information across the entire vocabulary at each reasoning step, our method allows the model to implicitly explore multiple reasoning paths in parallel, leading to both higher accuracy and greater token efficiency. Experiments on mathematical and coding benchmarks demonstrate that *Soft Thinking* consistently improves pass@1 accuracy and reduces generation length, all without any additional training or architectural modifications. Qualitative analyses further show that the reasoning process remains interpretable and concise. Future work may explore integrating training-based approaches to adapt the *concept token*, with the goal of further improving performance and stability when faced with out-of-distribution inputs.

## 6  Acknowledgments

We would like to express our sincere gratitude to Yue Fan, Saaket Agashe, Liliang Ren, Hao Cheng, Baolin Peng, and Yiping Wang for their valuable feedback and constructive discussions. We thank Orby AI for generously providing the computational resources. Additionally, we appreciate the SGLang Team's assistance during development.

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

# A Appendix

## A.1 Limitation

While *Soft Thinking* demonstrates significant improvements in both reasoning accuracy and token efficiency without requiring any additional training, its training-free nature also introduces certain limitations. Specifically, since current large language models are trained exclusively on discrete token sequences, they have never encountered *concept tokens*, the probability-weighted mixtures of token embeddings, during pre-training or fine-tuning. As a result, feeding such continuous *concept tokens* into the model during inference places it in an out-of-distribution (OOD) regime. This can lead to instability or generation collapse, especially when the reasoning chain becomes long or the input distribution diverges from the model's training data. Although our *Cold Stop* mechanism helps mitigate these issues by terminating reasoning when the model is sufficiently confident, it does not fundamentally resolve the OOD problem. Future work should explore training strategies that explicitly expose the model to *concept tokens*, thereby aligning its internal representations with the continuous concept space and improving robustness and generalization under soft thinking paradigms.

## A.2 Benchmarks

The evaluation covers four mathematical benchmark tasks and three programming benchmark tasks.

- **Mathematical Benchmarks:** Math500 [40] is a diverse subset of 500 problems selected from the MATH dataset [31], covering seven mathematical disciplines. AIME 2024 [32] is drawn from the American Invitational Mathematics Examination, this dataset features challenging problems that serve as a rigorous benchmark for both accuracy and token efficiency. GSM8K [33] is a benchmark comprising 1,319 grade-school-level math word problems designed to assess multi-step arithmetic reasoning. GPQA-Diamond [34] is a subset focusing on high-difficulty problems requiring deep, multi-step reasoning.

- **Coding Benchmarks:** HumanEval [35] is a widely used benchmark for evaluating functional correctness in code generation. Each problem consists of a Python function signature and a natural language description, with correctness measured by executing unit tests. MBPP [36] contains introductory-level coding tasks paired with simple specifications and measures a model's ability to understand and execute basic programming logic. LiveCodeBench [37] is a dynamic, contamination-free code generation benchmark. For our study, we select 279 problems released between August 2024 and January 2025 to ensure evaluation on unseen tasks.

## A.3 Implementation of *Soft Thinking* on SGLang

In this appendix, we describe the engineering modifications made to the SGLang inference engine (`v0.4.6.post1`) to support our proposed **Soft Thinking** method. We highlight the core code changes, affected files, and provide high-level code snippets to clarify the new reasoning flow.

### A.3.1 Overview of Modifications

Our implementation introduces a new inference mode, *soft thinking*, in which intermediate reasoning steps use "concept tokens" (i.e., full token probability distributions) rather than discrete token ids. This required changes to the input/output interface, sampling, embedding, and state management in SGLang. The main modifications are summarized as follows:

- **Configuration**: New flags and parameters to enable soft thinking and control its behavior.

- **Sampler**: Modified to output top-$k$ probability distributions (concept tokens) instead of a single sampled token.

- **Embedding Layer**: Added support for probability-weighted interpolation of token embeddings.

- **Forward Pipeline**: Adapted to accept and propagate concept tokens through the model.

- **Cold Stop**: Entropy-based early stopping logic for intermediate reasoning steps.

### A.3.2 Key Files and Logic Changes

**1.** `model_config.py` **&** `server_args.py`

- **Purpose**: Add configuration options for soft thinking.
- **Key additions**:

```
1  enable_soft_thinking: bool
2  max_topk: int
3  # Command-line flags: --enable-soft-thinking, --max-topk, --
        think-end-str
```

**2.** `sampler.py` **(Sampling Logic)**

- **Purpose**: Output concept tokens (top-$k$ probabilities and indices) instead of only discrete token ids.
- **Key changes**:

```
1  if enable_soft_thinking:
2      # Compute top-k probabilities and indices
3      topk_probs, topk_indices = torch.topk(probs, k=max_topk,
           dim=-1)
4      # Normalize
5      topk_probs = topk_probs / topk_probs.sum(dim=-1, keepdim=
           True)
6      logits_output.topk_probs = topk_probs
7      logits_output.topk_indices = topk_indices
8      # For next token id, use argmax or sample from topk as
           needed
9      batch_next_token_ids = topk_indices[:, 0]
10 else:
11     # Standard discrete sampling
12     batch_next_token_ids = torch.argmax(probs, -1)
13
14 # Entropy calculation for Cold Stop
15 entropy = -torch.sum(probs * torch.log(probs.clamp(min=1e-12)),
        dim=-1)
16 logits_output.entropy = entropy
```

**3.** `vocab_parallel_embedding.py` **(Embedding Layer)**

- **Purpose**: Support probability-weighted embedding computation.
- **Key changes**:

```
1  def weighted_forward(self, topk_probs, topk_indices):
2      # Compute weighted sum of embeddings
3      topk_embeddings = self.quant_method.embedding(self,
           topk_indices.long())   # [B, K, D]
4      # Normalize probabilities
5      topk_probs = topk_probs / topk_probs.sum(dim=-1, keepdim=
           True)
6      new_embedding = torch.sum(topk_probs.unsqueeze(-1) *
           topk_embeddings, dim=1)
7      return new_embedding
```

**4.** `models/llama.py`**,** `models/qwen2.py` **(Model Forward Pass)**

- **Purpose**: Accept and process concept tokens as input.
- **Key changes**:

```
1   # In model.forward:
2   if forward_batch.topk_probs is not None and forward_batch.
        topk_indices is not None:
3       if self.tp_size > 1:
4           hidden_states = self.embed_tokens.weighted_forward_tp(
                forward_batch.topk_probs, forward_batch.topk_indices
                )
5       else:
6           hidden_states = self.embed_tokens.weighted_forward(
                forward_batch.topk_probs, forward_batch.topk_indices
                )
7   elif input_embeds is None:
8       hidden_states = self.embed_tokens(input_ids)
```

**5.** `schedule_batch.py`, `scheduler.py`, `scheduler_output_processor_mixin.py`

- **Purpose**: State management and output tracking for soft thinking.
- **Key changes**:

```
1   # Pseudocode for Cold Stop
2   if entropy < entropy_threshold:
3       low_entropy_steps += 1
4   else:
5       low_entropy_steps = 0
6   if low_entropy_steps >= length_threshold:
7       # Insert end-of-thinking token, switch to answer mode
8       self.output_ids[-1] = self.sampling_params.think_end_str_id
```

**6.** `sampling_params.py`, `sampling_batch_info.py`

- **Purpose**: Add soft thinking-specific parameters and per-batch flags.
- **Key changes**:

```
1   # Parameters for after-thinking sampling, entropy thresholds,
        etc.
2   early_stopping_entropy_threshold: float
3   early_stopping_length_threshold: int
4   soft_thinking_mode: Optional[torch.Tensor]
```

### A.3.3  High-Level Soft Thinking Inference Flow

1. **Initialization**: If `enable_soft_thinking` is set, the model enters soft thinking mode for reasoning steps.

2. **At Each Reasoning Step**:
   - The sampler outputs a concept token: top-$k$ probabilities and indices (not just a single token id).
   - The embedding layer computes the next input embedding as a weighted sum over token embeddings, using these probabilities.
   - The model forward pass consumes this weighted embedding.

3. **Cold Stop (Early Termination)**:
   - At each step, compute the entropy of the concept token.
   - If entropy is below a threshold for several consecutive steps, insert the end-of-thinking token to terminate reasoning.

4. **Answer Generation**: After reasoning, the model switches back to standard discrete decoding for the answer.

The above changes enable SGLang to support soft thinking as described in our paper, allowing for continuous, distributional reasoning and entropy-based early stopping, all with minimal overhead to the standard inference pipeline.

