# OpenReview forum: "Soft Thinking: Unlocking the Reasoning Potential of LLMs in Continuous Concept Space"
_NeurIPS.cc/2025/Conference — NeurIPS 2025 poster_

### Official Review · Reviewer_txtk · 2025-06-30

**Clarity:** 3
**Significance:** 3
**Originality:** 2
**Rating:** 4
**Confidence:** 4

**Summary:**

This paper introduces Soft Thinking, a training-free method that enhances the reasoning capabilities of LLMs by operating in a continuous concept space. Instead of discrete token sampling (as in Chain-of-Thought), Soft Thinking generates concept tokens-probability-weighted mixtures of token embeddings- enabling implicit parallel exploration of reasoning paths. Key innovations include:
1. Concept tokens: Replace discrete tokens with continuous embeddings aggregated from the full vocabulary distribution.
2. Cold Stop: An entropy-based early termination mechanism to prevent generation collapse under out-of-distribution inputs.
The method improves Pass@1 accuracy by up to 2.48% and reduces token usage by up to 22.4% on math/coding benchmarks (e.g., MATH500, HumanEval) across models like Qwen-32B and DeepSeek-70B, while maintaining interpretability.

**Questions:**

1. OOD Robustness: The paper notes concept tokens place models in OOD regimes (Appendix A.1). How does performance degrade on highly OOD tasks (e.g., non-STEM)? Could lightweight finetuning on concept embeddings mitigate this?
2. Hyperparameter Sensitivity: Cold Stop uses thresholds τandk . How do results vary with these values (e.g., lowτcausing premature termination)? A sensitivity analysis would clarify robustness.
3. Scalability to Smaller Models: The method assumes decoupled input/output embeddings (Section 2.2), which is typical for >7B models. Would Soft Thinking work on sub-7B models (e.g., via weight tying)?

**Ethical Concerns:**

["NO or VERY MINOR ethics concerns only"]

**Final Justification:**

The authors paritially addressed my concerns. Since the original score I gave is positive, It is appropriate, so I will keep the score of 4.

**Limitations:**

Yes

**Quality:**

3

**Strengths And Weaknesses:**

• Strengths:
1. Model novelty
(1) Train a free continuous reasoning paradigm: Implement Probabilistic Embedding Propagation (PEP) by directly reusing pre trained word embedding layers (without adding new parameters) to transform discrete inference into a continuous space optimization problem.
(2) Lightweight Implementation of Implicit Multipath Exploration: By implicitly integrating multiple inference paths through the concept token e (k) in a single forward propagation, the computational complexity is only O (n⋅d) O (n⋅ d) (n≪∣V∣n≪∣V∣).
2. Rigorous evaluation across 8 benchmarks (4 math, 3 coding) and 3 model architectures (32B–70B).
3. Ablation studies validate design choices (e.g., Table 3: Soft Thinking vs. average embedding/COCONUT).
4. Theoretical grounding: Section 3.3 formally links concept tokens to path-summation approximation.

• Weaknesses:
1. Rationality analysis of model modeling
(1) Problem 1: Semantic collapse risk in conceptual space
The concept token ct (k) is essentially a convex combination of word embeddings, but linear interpolation in high-dimensional embedding spaces may not maintain semantic consistency (such as "king"+"woman" - "man" ≈ "queen" only holds in specific directions). Will weighted embedding lead to semantic ambiguity when p is evenly distributed across multiple unrelated concepts (such as activating "integral" and "loop" simultaneously in mathematical reasoning)?
(2) Problem 2: The boundary between probability decoding methods is blurred
The core operation of Soft Thinking - weighted embedding input - is highly similar to the following methods: COCONUT [26]: Use the hidden state h(t) as the next time input (but require training adapter); DoLa (Chuang et al. 2023, Decoding by Contrasting Layers): Weighted different layer representations generate concept vectors. This paper claims that 'no training is required', but does not prove that its mechanism is fundamentally different from the probability driven decoding mentioned above.
(3) The incremental innovation of Cold Stop
Entropy threshold cold stop is essentially a direct application of classical uncertainty control (such as [28]), and Not more efficient than Speculative Decoding (Leviathan et al. 2023) (the latter can skip multiple steps); The connection with Adaptive Computation Time (ACT) has not been discussed.
2. Limited analysis of failure cases (e.g., when Cold Stop truncates useful reasoning prematurely).
3. Benchmarks focus on STEM; generality to commonsense reasoning is untested.
4. Related Works to Discuss
(1) Comparative discussion: Clearly distinguish the difference in computational efficiency between ToT/GoT/Soft Prompt (e.g. ToT requires 10-100x LM calls, this article only makes a single call).
[1] Yao et al. (2023). Tree of Thoughts: Deliberate Problem Solving with Large Language Models. NeurIPS.
[2] Besta et al. (2024). Graph of Thoughts: Solving Elaborate Problems with Large Language Models. ICLR.
[3] He et al. (2024) Decoding by Contrasting Soft Prompts. NeurIPS.
(2) Enhance Theoretical analysis: The theoretical foundation supporting entropy threshold.
[4] Xin et al. (2021). BERxiT: Early Exiting for BERT with Better Fine-Tuning and Extension to Regression. EACL.
[5] Lin et al. (2024). Calibrating LLM Confidence via Bayesian Inference. ICML.
(3) Extended experiment: Test Soft Thinking on LeanDojo or PoT tasks to verify its generalization in formal reasoning.
[6] Chen et al. (2022). Program of Thoughts Prompting: Disentangling Computation from Reasoning. NeurIPS.
[7] Yang et al. (2023). LeanDojo: Theorem Proving with Retrieval-Augmented Language Models. NeurIPS.

---

> ### Author Rebuttal · Authors · 2025-07-29
>
> We sincerely appreciate your thoughtful and detailed feedback. Below, we provide thorough responses to your questions and concerns.
>
> ## **1. Question**
> ### **1.1 OOD Robustness**
> Soft Thinking does face OOD issues. **However, the OOD problem here does not refer to tasks or problems. Instead, it specifically refers to the fact that the weighted sum embeddings of concept tokens were not seen during training.**
> We believe lightweight fine-tuning could mitigate this issue. However, this work focuses on introducing the concept and algorithm itself, leaving robustness optimization as future work.
>
> ## **1.2 Sensitivity Analysis**
> We conducted hyperparameter sensitivity analysis on QwQ-32B and DeepSeek-Distill-Qwen-32B across AIME2024 and LiveCodeBench datasets.
> ### **1.2.1 QwQ32b (aime2024)**
> #### **Length = 256**
> | Entropy | 0 | 0.01  | 0.05  | 0.075 | 0.1 | 0.2 | 0.3 | 0.5 |
> | - | - | - | - | - | - | - | - | - |
> | Accuracy (%) | 73.33 | 80.00    | 76.66 | 83.33 | 83.33 | 83.33 | 83.33 | 80.00    |
> | Tokens | 9457  | 11428 | 9910  | 10537 | 10514 | 11045 | 11873 | 10611 |
>
> #### **Entropy = 0.1**
> | Length | 64   | 128   | 256   | 512   | 1024  |
> | - | - | - | - | - | - |
> | Accuracy (%) | 80.00   | 80.00    | 83.33 | 83.33 | 73.33 |
> | Tokens | 9933 | 10433 | 10514 | 11228 | 9027  |
> ---
> ### **1.2.2 QwQ-32B (livecodebench)**
> #### **Length = 512**
> | Entropy | 0 | 0.01  | 0.05  | 0.075 | 0.1 | 0.2 | 0.3 | 0.5 |
> |-|-|-|-|-|-|-|-|-|
> | Accuracy (%) | 60.93 | 62.72 | 62.72 | 62.72 | 62.72 | 62.36 | 60.93 | 60.93 |
> | Tokens | 7378  | 7564  | 7535  | 7463  | 7808  | 7591  | 7630  | 7295  |
>
> #### **Entropy = 0.05**
> | Length | 64 | 128 | 256 | 512 | 1024 |
> |-|-|-|-|-|-|
> | Accuracy (%) | 60.21 | 60.21 | 62.36 | 62.72 | 62.72 |
> | Tokens | 7521  | 7652  | 7711  | 7535  | 7820  |
> ---
> ### **1.2.3 DeepSeek-R1-Distill-Qwen-32B (aime2024)**
> #### **Length=256**
> | Entropy | 0 | 0.01  | 0.05  | 0.075 | 0.1 | 0.2 | 0.3 | 0.5  |
> |-|-|-|-|-|-|-|-|-|
> | Accuracy (%) | 70 | 76.66 | 73.33 | 76.66 | 76.66 | 73.33 | 73.33 | 70   |
> | Tokens | 5923 | 6497  | 6275  | 6533  | 6620  | 6377  | 6462  | 6531 |
>
> #### **Entropy = 0.1**
> | Length | 64 | 128 | 256 | 512 | 1024  |
> |-|-|-|-|-|-|
> | Accuracy (%) | 70 | 76.66 | 76.66 | 76.66 | 73.33 |
> | Tokens | 5890 | 6543  | 6620  | 6888  | 6457  |
> ---
> ### **1.2.4 DeepSeek-R1-Distill-Qwen-32B (livecodebench)**
> #### **Length=512**
> | Entropy      | 0     | 0.01 | 0.05 | 0.075 | 0.1  | 0.2   | 0.3   | 0.5   |
> |-|-|-|-|-|-|-|-|-|
> | Accuracy (%) | 56.98 | 59.5 | 59.5 | 59.5  | 59.5 | 58.78 | 58.42 | 56.63 |
> | Tokens | 6877  | 6351 | 6275 | 6199  | 6335 | 6419  | 6253  | 6127  |
>
> #### **Entropy = 0.05**
> | Length | 64 | 128 | 256 | 512  | 1024 |
> |-|-|-|-|-|-|
> | Accuracy (%) | 53.76 | 56.63 | 58.78 | 59.5 | 59.5 |
> | Tokens | 6935  | 6139  | 6213  | 6255 | 6402 |
>
> ### **1.2.5 Analysis**
> The results demonstrate that our method is robust to hyperparameter variations across a large spectrum:
> - **Entropy Sensitivity**:
>   For QwQ-32B on AIME2024, entropy threshold ranging from 0.075 to 0.3 consistently achieve optimal results, exceeding the baseline. Similarly, for LiveCodeBench, entropy threshold values between 0.01 and 0.2 outperform the baseline. For DeepSeek-32B, the trends are similar, showing robustness across varying entropy thresholds.
> - **Length Sensitivity**:
>   On AIME2024, lengths threshold from 64 to 512 work effectively, while on LiveCodeBench, lengths threshold from 256 to 1024 exceed baseline performance.
>
> Optimal hyperparameter ranges vary slightly across tasks but share a large overlap and are independent of the models used. Coding tasks require lower entropy and longer lengths threthod to generate longer code blocks.
>
> **For general settings**, entropy = 0.1 and length = 512 perform well across all datasets and models, consistently improving upon the baseline. In our paper, for optimal results, we selected entropy = 0.1 and length = 256 for math tasks, and entropy = 0.05 and length = 512 for code tasks.
>
> **Therefore, our method has good robustness to parameters and does not require heavy parameter tuning.**
>
> We will add these important sensitive analysis to camera ready version.
>
> ## **1.3. Smaller Models**
> Additional experiments on Deepseek-distill-qwen series models:
> | Model Size | Method | Pass@1 Accuracy (%) | Token Usage |
> |-|-|-|-|
> | 7B| CoT | 55.00 | 10,502 |
> |  | Soft Thinking  | 49.17 | 15,479 |
> | 14B  | CoT | 63.33 | 9,562 |
> |  | Soft Thinking  | 63.33 | 7,705 |
> | 32B | CoT | 72.08 | 9,347 |
> | | Soft Thinking  | 76.66 | 6,620 |
>
> Our experiments reveal that Soft Thinking does not significantly outperform the baseline on smaller models. However, **as model size increases, Soft Thinking gradually surpasses the baseline.** This is due to the following reasons:
> 1. In smaller models, even embeddings of semantically distant tokens may not form angles close to 90°, resulting in a higher proportion of irrelevant tokens in the probability distribution and increased noise.
> 2. Concept tokens are OOD, requiring the model to possess strong generalization abilities to understand embeddings derived from linear interpolation.
>
> Moreover, our work addresses the issue faced by methods like COCONUT, which rely heavily on weight tying and therefore fail to work effectively on larger models.
>
> ## **2. Weakness**
> ### **2.1 Problem 1: Semantic Collapse Risk**
> Thank you for your insightful observation. To clarify, Soft Thinking does not aggregate embeddings of tokens with vastly different semantics. The weighted probabilities are derived from the hidden state multiplied by the LM head and processed through a softmax function. This ensures that tokens with higher weights are semantically closer to the hidden state, while tokens with distant or opposite semantics have near-zero probabilities in the distribution.
>
> For instance, combinations like "I'll start" and "can use" at positions 13 and 14 (as shown in Figure 4) represent semantically consistent reasoning paths. In contrast, unrelated concepts such as "integral" and "loop" would not appear simultaneously in the same token. We hope this explanation alleviates concerns about semantic collapse risk. We will include this clarification in the revised paper. Thank you for pointing it out.
>
> ### **2.2 Problem 2: Similarity to Other Methods**
> We want to clarify that our approach is fundamentally different from these works. Soft Thinking innovatively extends LLM reasoning into continuous space without requiring additional modules or training. Below are specific distinctions:
>
> - **COCONUT**: Our method overcomes several limitations of COCONUT, including reliance on weight tying, the need for training, and incompatibility with large models. These differences were discussed in the related work section (line 92-100).
>
> - **DoLA**: The statement **Weighted different layer representations generate concept vectors** is not very accurate. DoLA does not use weighted representations. Instead, it contrasts the probability distributions of premature layers and mature layers. In addition, DoLA amplifies factual knowledge encoded in higher layers by contrasting probabilities but does not generate abstract "concept vectors and still use discrete tokens.
>
> Nevertheless, thank your for pointing out these methods. We will discuss them in camera ready version for clarification.
>
> ---
>
> ### **2.3 Cold Stop**
> Cold Stop is a straightforward and effective approach to reducing token counts and addressing OOD issues. We primarily validate the effectiveness of Cold Stop through experiments detailed in the ablation section. Regarding the theoretical aspect, due to space constraints, we will provide a more comprehensive theoretical derivation in the camera-ready version.
>
> ---
>
> ### **2.4 Comparison to Speculative Decoding and ACT**
> Thank you for sharing these methods. Speculative Decoding focuses on accelerating decoding speed by using smaller models, not skipping several tokens. ACT determines the number of computational steps or layers needed based on input complexity. While ACT operates vertically (layer-wise), Soft Thinking operates horizontally (reasoning steps). These are orthogonal optimization directions, and Soft Thinking could be combined with these methods, which we consider a promising future direction. These possible research directions don't impact the innovation of Soft Thinking.
>
> ---
>
> ### **2.5 Limited Analysis of Failure Cases**
> Our experiments reveal that tasks requiring lengthy code generation demand higher thresholds, as lower ones may cause premature termination. We will include examples illustrating this issue in the revised paper.
>
> ---
>
> ### **2.6 Benchmarks Focus on STEM**
> The seven benchmarks (four math, three coding) we selected are widely recognized and commonly used, providing a comprehensive comparison of different methods. However, we will expand our experiments to include additional benchmarks in the revised paper to further validate generalization.
>
> ---
> ### **2.7 Related Works Comparison**
> Methods like Tree of Thoughts (ToT) and Graph of Thoughts (GoT) require extensive sampling (e.g., 16-64 samples for self-consistency), making their token usage significantly higher (up to 64x CoT). We will add experiments in the revised paper to highlight the trade-offs between effectiveness and efficiency, due to the constraints of the rebuttal period and computational resources.
>
> Regarding Soft Prompt, while both methods contain "soft" in their names, they are fundamentally different. Soft Prompt focuses on parameter-efficient fine-tuning, requiring additional training and fixed soft prompts during inference. In contrast, Soft Thinking expands reasoning tokens into continuous space, addressing a different problem altogether.
>
> # **Acknowledgment**
> We sincerely respect and appreciate the hard work and professionalism demonstrated in your review. We look forward to your feedback on our revisions and thank you again for your valuable time and effort.

---

> > ### Comment · Reviewer_txtk · 2025-08-05
> >
> > Thank the authors for the rebuttal. The response partially addressed my concerns, but I believe the original score is appropriate, so I will keep the score.

---

> ### Comment · Area_Chair_EYC9 · 2025-08-04
> **Does the response address your concerns?**
>
> Hi, when you have a moment, could you please take a look at the authors' response to see whether it addresses your concerns, or if there are still points that remain unclear? Thank you.

---

> ### Author Response · Authors · 2025-08-06
> **Thank you for the response**
>
> Thank you for acknowledging that our responses have addressed your previous concerns. We appreciate your recognition of our work.
>
> While our response has partially addressed them, we would like to kindly ask for clarification on which specific aspects remain unresolved. We are more than happy to discuss further and provide additional explanations if needed.
>
> Best,
>
> Authors

---

### Official Review · Reviewer_81RM · 2025-07-01

**Clarity:** 3
**Significance:** 2
**Originality:** 3
**Rating:** 4
**Confidence:** 2

**Summary:**

The paper introduces "Soft Thinking," a training-free method enabling LLMs to reason in a continuous concept space by replacing discrete tokens with probability-weighted concept tokens. This approach allows models to explore multiple reasoning paths implicitly, improving accuracy and reducing token usage. Experiments on math and coding benchmarks show up to 2.48% higher pass@1 accuracy and 22.4% shorter generation lengths compared to standard CoT. The Cold Stop mechanism mitigates generation collapse, and theoretical analysis frames Soft Thinking as a linear approximation of path-summation in CoT.

**Questions:**

see weakness.

**Ethical Concerns:**

["NO or VERY MINOR ethics concerns only"]

**Final Justification:**

The author's reply has basically addressed my concern, and I tend to keep my postive score (4:Borderline accept).

**Limitations:**

yes

**Quality:**

2

**Strengths And Weaknesses:**

- The text in the image is too small, which is unfriendly to readers.
- Does Soft Thinking struggle with extremely long reasoning tasks where out-of-distribution (OOD) inputs accumulate?
- The entropy threshold (τ) and consecutive steps (k) are fixed hyperparameters. For example, using τ=0.01 and k=128 (Section 4.2, lines 242–243) may not generalize to diverse problem distributions, potentially leading to premature termination or redundant steps. How to mitigate and discuss this issue?
- On LiveCodeBench, Cold Stop reduces generation length but increases the correct solution length (Table 3, lines 325–326), indicating that the stopping conditions for complex code tasks may be suboptimal. Is my understanding correct?
- The paper compares against standard and greedy Chain of Thought (CoT) but lacks benchmarks against state-of-the-art methods like Tree of Thoughts or self-consistency, which also explore multiple reasoning paths.

---

> ### Author Rebuttal · Authors · 2025-07-29
>
> We sincerely appreciate your thoughtful and detailed feedback. Below, we provide thorough responses to your questions and concerns.
>
> ## **1. The text in the image is too small.**
> Thank you for your suggestion. We will adjust the proportions between the blank space and the text in the images, enlarge the text, and enhance the contrast between the text and the background to improve readability.
>
> ## **2. Extremely Long OOD Inputs**
> Soft Thinking is capable of handling extremely long OOD inputs effectively. For instance, on the AIME2024 dataset, the average reasoning length exceeds 10k tokens, while on LiveCodeBench, it surpasses 7k tokens, with the maximum length reaching over 30k tokens. These results demonstrate that the model performs well on extremely long reasoning tasks.
>
> ## **3. Hyperparameter Sensitivity Analysis**
> We conducted hyperparameter sensitivity analysis on QwQ-32B and DeepSeek-Distill-Qwen-32B across AIME2024 and LiveCodeBench datasets.
> ### **3.1 QwQ32b (aime2024)**
>
> #### **Length = 256, Varying Entropy**
>
> | Entropy      | 0     | 0.01  | 0.05  | 0.075 | 0.1   | 0.2   | 0.3   | 0.5   |
> | ------------ | ----- | ----- | ----- | ----- | ----- | ----- | ----- | ----- |
> | Accuracy (%) | 73.33 | 80.00    | 76.66 | 83.33 | 83.33 | 83.33 | 83.33 | 80.00    |
> | Tokens       | 9457  | 11428 | 9910  | 10537 | 10514 | 11045 | 11873 | 10611 |
>
> #### **Entropy = 0.1, Varying Length**
>
> | Length       | 64   | 128   | 256   | 512   | 1024  |
> | ------------ | ---- | ----- | ----- | ----- | ----- |
> | Accuracy (%) | 80.00   | 80.00    | 83.33 | 83.33 | 73.33 |
> | Tokens       | 9933 | 10433 | 10514 | 11228 | 9027  |
>
> ---
>
> ### **3.2 QwQ-32B (livecodebench)**
>
> #### **Length = 512, Varying Entropy**
>
> | Entropy      | 0     | 0.01  | 0.05  | 0.075 | 0.1   | 0.2   | 0.3   | 0.5   |
> | ------------ | ----- | ----- | ----- | ----- | ----- | ----- | ----- | ----- |
> | Accuracy (%) | 60.93 | 62.72 | 62.72 | 62.72 | 62.72 | 62.36 | 60.93 | 60.93 |
> | Tokens       | 7378  | 7564  | 7535  | 7463  | 7808  | 7591  | 7630  | 7295  |
>
> #### **Entropy = 0.05, Varying Length**
>
> | Length       | 64    | 128   | 256   | 512   | 1024  |
> | ------------ | ----- | ----- | ----- | ----- | ----- |
> | Accuracy (%) | 60.21 | 60.21 | 62.36 | 62.72 | 62.72 |
> | Tokens       | 7521  | 7652  | 7711  | 7535  | 7820  |
>
> ---
>
> ### **3.3 DeepSeek-R1-Distill-Qwen-32B (aime2024)**
>
> #### **Length=256, Varying Entropy**
>
> | Entropy      | 0    | 0.01  | 0.05  | 0.075 | 0.1   | 0.2   | 0.3   | 0.5  |
> | ------------ | ---- | ----- | ----- | ----- | ----- | ----- | ----- | ---- |
> | Accuracy (%) | 70   | 76.66 | 73.33 | 76.66 | 76.66 | 73.33 | 73.33 | 70   |
> | Tokens       | 5923 | 6497  | 6275  | 6533  | 6620  | 6377  | 6462  | 6531 |
>
> #### **Entropy = 0.1, Varying Length**
>
> | Length       | 64   | 128   | 256   | 512   | 1024  |
> | ------------ | ---- | ----- | ----- | ----- | ----- |
> | Accuracy (%) | 70   | 76.66 | 76.66 | 76.66 | 73.33 |
> | Tokens       | 5890 | 6543  | 6620  | 6888  | 6457  |
>
> ---
>
> ### **3.4 DeepSeek-R1-Distill-Qwen-32B (livecodebench)**
>
> #### **Length=512, Varying Entropy**
>
> | Entropy      | 0     | 0.01 | 0.05 | 0.075 | 0.1  | 0.2   | 0.3   | 0.5   |
> | ------------ | ----- | ---- | ---- | ----- | ---- | ----- | ----- | ----- |
> | Accuracy (%) | 56.98 | 59.5 | 59.5 | 59.5  | 59.5 | 58.78 | 58.42 | 56.63 |
> | Tokens       | 6877  | 6351 | 6275 | 6199  | 6335 | 6419  | 6253  | 6127  |
>
> #### **Entropy = 0.05, Varying Length**
>
> | Length       | 64    | 128   | 256   | 512  | 1024 |
> | ------------ | ----- | ----- | ----- | ---- | ---- |
> | Accuracy (%) | 53.76 | 56.63 | 58.78 | 59.5 | 59.5 |
> | Tokens       | 6935  | 6139  | 6213  | 6255 | 6402 |
>
> ### **3.5 Analysis**
> The results demonstrate that our method is robust to hyperparameter variations across a large spectrum:
>
> - **Entropy Sensitivity**:
>   For QwQ-32B on AIME2024, entropy threshold values ranging from 0.075 to 0.3 consistently achieve optimal results, exceeding the baseline. Similarly, for LiveCodeBench, entropy threshold values between 0.01 and 0.2 outperform the baseline. For DeepSeek-32B, the trends are similar, showing robustness across varying entropy thresholds.
>
> - **Length Sensitivity**:
>   On AIME2024, lengths threshold from 64 to 512 work effectively, while on LiveCodeBench, lengths threshold from 256 to 1024 exceed baseline performance.
>
> Optimal hyperparameter ranges vary slightly across tasks but share a large overlap and are independent of the models used. Coding tasks require lower entropy and longer lengths threthod to generate longer code blocks, as overly aggressive stopping can truncate meaningful outputs.
>
> **For general settings**, entropy = 0.1 and length = 512 perform well across all datasets and models, consistently improving upon the baseline. In our paper, for optimal results, we selected entropy = 0.1 and length = 256 for math tasks, and entropy = 0.05 and length = 512 for code tasks.
>
> **Therefore, our method has good robustness to parameters and does not require heavy parameter tuning.**
>
> We will add these important sensitive analysis to camera ready version.
>
>
>
> ## **4. Cold Stop and generation length on LiveCodeBench:**
>
> The increase in average generation length for correct solutions with Cold Stop is a result of the model solving more problems, many of which require longer reasoning chains. Specifically, without Cold Stop, the model achieves an accuracy of 56.98%, with an average generation length of 6,877 tokens for correct solutions. When Cold Stop is applied, the accuracy improves significantly to 62.72%, allowing the model to correctly solve more problems, including many that demand reasoning chains exceeding 10k tokens.
>
> As a result, the inclusion of these more challenging problems naturally increases the average generation length for correct solutions to 7,535 tokens. This does not indicate that Cold Stop's stopping conditions are suboptimal for complex code tasks. Instead, it highlights Cold Stop helps to handle more difficult problems, which require longer reasoning chains.
>
> ## **5. Comparison with Tree of Thoughts and self-consistency:**
> Your suggestion regarding the lack of comparison with these methods is very valid. The primary reason is that our approach targets both effectiveness and efficiency, while methods like Tree of Thoughts or self-consistency require extensive sampling or searches. For example, self-consistency often requires sampling 16 or 32 times, leading to token usage that is 16 or 32 times higher than the baseline CoT. Moreover, integrating ToT and self-consistency into SGLang demands significant development time. Given the constraints of the rebuttal period and computational resources, we will conduct experiments comparing these methods in the camera-ready version to emphasize the trade-offs between effectiveness and efficiency.
>
> ## **Acknowledgment**
> We genuinely appreciate the diligent and professional nature of your review. Your insightful comments have been invaluable in refining our paper, and we trust that the clarifications above adequately resolve your concerns. We eagerly await your response to our revisions and extend our gratitude once more for your valuable time and effort.

---

> > ### Comment · Reviewer_81RM · 2025-08-07
> >
> > Thank you for your detailed response. The author's reply has basically addressed my concern. I am currently considering the suggestions from other reviewers, especially for the reviewer zaBE, and I tend to keep my postive score first.

---

> ### Comment · Area_Chair_EYC9 · 2025-08-04
> **Does the response address your concerns?**
>
> Hi, when you have a moment, could you please take a look at the authors' response to see whether it addresses your concerns, or if there are still points that remain unclear? Thank you.

---

> > ### Author Response · Authors · 2025-08-06
> >
> > Dear Reviewer 81RM,
> >
> > Thank you once again for your valuable comments on our submission. As the discussion phase is approaching its end, we would like to kindly confirm whether we have sufficiently addressed your concerns. If there are any remaining questions or areas requiring further clarification, please do not hesitate to let us know. Your suggestions have been invaluable in improving our paper, and we would greatly appreciate your reply.
> >
> > Thank you for your time and constructive feedback.
> >
> > Sincerely,
> >
> > Authors

---

### Official Review · Reviewer_m4ot · 2025-07-02

**Clarity:** 3
**Significance:** 3
**Originality:** 3
**Rating:** 4
**Confidence:** 3

**Summary:**

This paper proposes a training-free reasoning approach for large language models that conducts reasoning in the continuous latent space rather than at the discrete token level (in contrast to traditional Chain-of-Thought reasoning). Through comprehensive experiments across multiple benchmarks, the authors demonstrate performance improvements using this continuous space reasoning methodology. The approach aims to leverage the representational richness of the latent space to enhance reasoning capabilities without requiring additional model training.

**Questions:**

Please refer to the Weaknesses section above

**Ethical Concerns:**

["NO or VERY MINOR ethics concerns only"]

**Final Justification:**

I still maintain my original score which has indicated an "accept"

**Limitations:**

Please refer to the Weaknesses section above

**Paper Formatting Concerns:**

No significant formatting issues identified.

**Quality:**

3

**Strengths And Weaknesses:**

Strengths:

- Proposes a training-free reasoning approach that operates in the continuous latent space rather than discrete token space, offering a fundamentally different perspective from traditional Chain-of-Thought (CoT) methods
- Clear methodological exposition: Provides a clear explanation of how the method differs from original CoT and specifically describes how reasoning is conducted in the continuous latent space. The paper also includes detailed theoretical derivations that support the method design and provide mathematical justification for the approach


Weaknesses:

- The paper claims that "the method preserves the uncertainty and richness inherent in reasoning by aggregating over all possible next steps," suggesting that soft thinking can overcome the limited sampling path diversity of traditional CoT. However, the explanation of how continuous latent space sampling enables token diversity and the ability to "explore multiple reasoning paths in parallel, rather than being limited to a single trajectory" requires further clarification. The mechanism by which this parallel exploration occurs is not convincingly demonstrated.

---

> ### Author Rebuttal · Authors · 2025-07-29
>
> We truly value the your comprehensive and insightful feedback. Below, we offer detailed replies to your questions.
>
> ## **Explanation to Parallel Thinking**
> ### **1. From a qualitative perspective**
> Soft Thinking retains the original probability distribution, preserving more information compared to traditional sampling methods. In standard sampling, the probability distribution collapses into a one-hot vector (where the position corresponding to the selected token is set to 1). This increases certainty, as the entropy of a one-hot distribution is zero, leaving no room for uncertainty.
>
> ### **2. From a theoretical perspective**
> Soft Thinking also provides a theoretical explanation for how it enables parallel reasoning.
>
> In Section 3.3, Equation (10) demonstrates:
>
> $$
> p(y \mid x) = \sum_{t_1} p(t_1 \mid x) p(y \mid x, t_1) \approx p\left(y \mid x, \sum_{t_1} p(t_1 \mid x) t_1 \right) = p(y \mid x, ct_1). \tag{10}
> $$
>
> Here, we prove that the concept token is a linear approximation of all possible discrete tokens at this step. The full outer summation over the first token is replaced by a single evaluation at the *concept token* $ct_1$ via linear approximation.
>
> Subsequently, in Equations (11) and (12), we apply the same linearization recursively:
>
> Given \( ct_1 \), the conditional probability expands as:
>
> $$
> p(y \mid x, ct_1) = \sum_{t_2} p(t_2 \mid x, ct_1) p(y \mid x, ct_1, t_2) \approx p(y \mid x, ct_1, ct_2), \tag{11}
> $$
>
> where $ct_2 = \sum_{t_2} p(t_2 \mid x, ct_1) t_2$.
>
> Repeating this process for all $m$ steps yields the continuous expansion:
>
> $$
> p(y \mid x) \approx p(y \mid x, ct_1, ct_2, \ldots, ct_m). \tag{12}
> $$
>
>
> We demonstrate that **the multi-step concept tokens ($ct_1, \ldots, ct_m$) are linear approximations of all possible path combinations of discrete tokens ($t_1, \ldots, t_m$) at each step**, weighted by their respective path probabilities. As a result, **a single reasoning path composed of concept tokens can approximate all possible discrete token reasoning paths, enabling the exploration of multiple reasoning paths in parallel within a single reasoning trajectory**.
>
> ### **3. Specific Examples**
> Additionally, as illustrated in Figure 4 of the paper, we observe examples of parallel reasoning in practice. For instance, in steps 13–14, combinations such as "I’ll start," "I can use," and "I’ll break" emerge; similarly, in steps 18–20, combinations like "break down the number into" and "break down the multiplication to" appear. These practical examples further demonstrate the ability of Soft Thinking to conduct parallel reasoning.
>
> Thus, the paper already provides qualitative evidence, theoretical proof, and case studies to explain how continuous space reasoning enables token diversity and facilitates the exploration of multiple reasoning paths in parallel.
>
> Nevertheless, we acknowledge the need for further clarity in this explanation. We will rewrite and improve this section in the revision to address the concerns raised by the reviewer.
>
> ## **Acknowledgment**
> We genuinely appreciate the diligent and professional nature of your review. Your insightful comments have been invaluable in refining our paper, and we trust that the clarifications offered above adequately address your concerns. We eagerly await your response to our updated submission and thank you once more for your considerable time and effort.

---

> ### Comment · Area_Chair_EYC9 · 2025-08-04
> **Does the response address your concerns?**
>
> Hi, when you have a moment, could you please take a look at the authors' response to see whether it addresses your concerns, or if there are still points that remain unclear? Thank you.

---

> > ### Author Response · Authors · 2025-08-06
> >
> > Dear Reviewer m4ot,
> >
> > Thank you once again for your valuable comments on our submission. As the discussion phase is approaching its end, we would like to kindly confirm whether we have sufficiently addressed your concerns. If there are any remaining questions or areas requiring further clarification, please do not hesitate to let us know. Your suggestions have been invaluable in improving our paper, and we would greatly appreciate your reply.
> >
> > Thank you for your time and constructive feedback.
> >
> > Sincerely,
> >
> > Authors

---

> > > ### Comment · Reviewer_m4ot · 2025-08-06
> > >
> > > Thank you for the authors' detailed response. Since I have already given a positive score leaning toward acceptance, I will maintain my original rating.

---

### Official Review · Reviewer_zaBE · 2025-07-03

**Clarity:** 3
**Significance:** 3
**Originality:** 3
**Rating:** 5
**Confidence:** 3

**Summary:**

This paper proposes to modify the inference process in generating CoT tokens by using a soft mix of embeddings instead of sampling the next token and then embedding it. The mixture weights are based on the output distribution, but with top-p and top-k constraints and renormalization. In order to make it work in practice, this work also introduced "Cold Stop", where when the entropy is lower than a fixed threshold for a set number of steps, they stopped using soft embeddings and revert back to normal decoding. Experiments on multiple recent state-of-the-art language models show that the proposed approach slightly improves reasoning accuracy with lower inference cost.

**Questions:**

Please see weaknesses.

**Ethical Concerns:**

["NO or VERY MINOR ethics concerns only"]

**Final Justification:**

Apologies for my delay. The author response does address my primary concerns:

1. Regarding how sensitive the method is with respect to hyper-parameters, the added experiments fully address my concerns.

2. Regarding cold stop, after reading authors' explanations, I think I misunderstood the method in my initial review.

Given this, I'm increasing my rating to 5.

**Limitations:**

Yes

**Paper Formatting Concerns:**

No, although at line 290 "Visualization of Shorted Example" the word "shorted" feels a bit weird. Does it mean soft thinking has shorter chains, or that part of the chain was abbreviated and not shown in the figure?

**Quality:**

3

**Strengths And Weaknesses:**

Strengths:
1. The proposed method is training-free and does not require finetuning the model.
2. The experiment results show that the proposed method can improve accuracy while using fewer CoT tokens for reasoning.

Weaknesses:
1. My primary concern is how practical the method is, due to the many hyperparameters introduced. For example, there's top-p, top-k, top-n, entropy threshold $\tau$, and length threshold $k$. This paper stated that "Results are reported based on the best-performing combinations of $\tau$ and $k$", but without a sensitivity analysis, it is hard to know whether we can use this method in practice. For example, if sensitivity analysis found that the improvements are observed across a large spectrum of hyperparameters, this method could be adapted in all language model decoding implementations, which would be a significant contribution (and I'd significantly increase my score); on the other hand, if it only works for specific combinations of hyperparameters (or the hyperparameters heavily depends on the dataset and the model), then the performance gains might more likely come from hyperparameter tuning.
2. Table 3 shows that Cold Stop significantly improves performance of soft thinking, but why not directly apply cold stop to the baseline normal decoding? Does the length reduction (or even accuracy improvement) come from cold stop itself?
3. Some related works in "Continuous Space Reasoning" are missing, such as https://arxiv.org/abs/2311.01460.

---

> ### Author Rebuttal · Authors · 2025-07-29
>
> We sincerely appreciate the reviewer’s thorough and constructive feedback on our paper. Your valuable insights have helped us identify areas for improvement and refine our work further. Below, we provide detailed responses to all the questions and concerns raised.
>
> ---
>
> ## **1. Hyperparameter Sensitivity Analysis**
> We conducted hyperparameter sensitivity analysis on QwQ-32B and DeepSeek-Distill-Qwen-32B across AIME2024 and LiveCodeBench datasets.
> ### **1.1 QwQ32b (aime2024)**
>
> #### **Length = 256, Varying Entropy**
>
> | Entropy      | 0     | 0.01  | 0.05  | 0.075 | 0.1   | 0.2   | 0.3   | 0.5   |
> | ------------ | ----- | ----- | ----- | ----- | ----- | ----- | ----- | ----- |
> | Accuracy (%) | 73.33 | 80.00    | 76.66 | 83.33 | 83.33 | 83.33 | 83.33 | 80.00    |
> | Tokens       | 9457  | 11428 | 9910  | 10537 | 10514 | 11045 | 11873 | 10611 |
>
> #### **Entropy = 0.1, Varying Length**
>
> | Length       | 64   | 128   | 256   | 512   | 1024  |
> | ------------ | ---- | ----- | ----- | ----- | ----- |
> | Accuracy (%) | 80.00   | 80.00    | 83.33 | 83.33 | 73.33 |
> | Tokens       | 9933 | 10433 | 10514 | 11228 | 9027  |
>
> ---
>
> ### **1.2 QwQ-32B (livecodebench)**
>
> #### **Length = 512, Varying Entropy**
>
> | Entropy      | 0     | 0.01  | 0.05  | 0.075 | 0.1   | 0.2   | 0.3   | 0.5   |
> | ------------ | ----- | ----- | ----- | ----- | ----- | ----- | ----- | ----- |
> | Accuracy (%) | 60.93 | 62.72 | 62.72 | 62.72 | 62.72 | 62.36 | 60.93 | 60.93 |
> | Tokens       | 7378  | 7564  | 7535  | 7463  | 7808  | 7591  | 7630  | 7295  |
>
> #### **Entropy = 0.05, Varying Length**
>
> | Length       | 64    | 128   | 256   | 512   | 1024  |
> | ------------ | ----- | ----- | ----- | ----- | ----- |
> | Accuracy (%) | 60.21 | 60.21 | 62.36 | 62.72 | 62.72 |
> | Tokens       | 7521  | 7652  | 7711  | 7535  | 7820  |
>
> ---
>
> ### **1.3 DeepSeek-R1-Distill-Qwen-32B (aime2024)**
>
> #### **Length=256, Varying Entropy**
>
> | Entropy      | 0    | 0.01  | 0.05  | 0.075 | 0.1   | 0.2   | 0.3   | 0.5  |
> | ------------ | ---- | ----- | ----- | ----- | ----- | ----- | ----- | ---- |
> | Accuracy (%) | 70   | 76.66 | 73.33 | 76.66 | 76.66 | 73.33 | 73.33 | 70   |
> | Tokens       | 5923 | 6497  | 6275  | 6533  | 6620  | 6377  | 6462  | 6531 |
>
> #### **Entropy = 0.1, Varying Length**
>
> | Length       | 64   | 128   | 256   | 512   | 1024  |
> | ------------ | ---- | ----- | ----- | ----- | ----- |
> | Accuracy (%) | 70   | 76.66 | 76.66 | 76.66 | 73.33 |
> | Tokens       | 5890 | 6543  | 6620  | 6888  | 6457  |
>
> ---
>
> ### **1.4 DeepSeek-R1-Distill-Qwen-32B (livecodebench)**
>
> #### **Length=512, Varying Entropy**
>
> | Entropy      | 0     | 0.01 | 0.05 | 0.075 | 0.1  | 0.2   | 0.3   | 0.5   |
> | ------------ | ----- | ---- | ---- | ----- | ---- | ----- | ----- | ----- |
> | Accuracy (%) | 56.98 | 59.5 | 59.5 | 59.5  | 59.5 | 58.78 | 58.42 | 56.63 |
> | Tokens       | 6877  | 6351 | 6275 | 6199  | 6335 | 6419  | 6253  | 6127  |
>
> #### **Entropy = 0.05, Varying Length**
>
> | Length       | 64    | 128   | 256   | 512  | 1024 |
> | ------------ | ----- | ----- | ----- | ---- | ---- |
> | Accuracy (%) | 53.76 | 56.63 | 58.78 | 59.5 | 59.5 |
> | Tokens       | 6935  | 6139  | 6213  | 6255 | 6402 |
>
> ### **1.5 Analysis**
> The results demonstrate that our method is robust to hyperparameter variations across a large spectrum:
>
> - **Entropy Sensitivity**:
>   For QwQ-32B on AIME2024, entropy threshold values ranging from 0.075 to 0.3 consistently achieve optimal results, exceeding the baseline. Similarly, for LiveCodeBench, entropy threshold values between 0.01 and 0.2 outperform the baseline. For DeepSeek-32B, the trends are similar, showing robustness across varying entropy thresholds.
>
> - **Length Sensitivity**:
>   On AIME2024, lengths threshold from 64 to 512 work effectively, while on LiveCodeBench, lengths threshold from 256 to 1024 exceed baseline performance.
>
> Optimal hyperparameter ranges vary slightly across tasks but share a large overlap and are independent of the models used. Coding tasks require lower entropy and longer lengths threthod to generate longer code blocks, as overly aggressive stopping can truncate meaningful outputs.
>
> **For general settings**, entropy = 0.1 and length = 512 perform well across all datasets and models, consistently improving upon the baseline. In our paper, for optimal results, we selected entropy = 0.1 and length = 256 for math tasks, and entropy = 0.05 and length = 512 for code tasks.
>
> **Therefore, our method has good robustness to parameters and does not require heavy parameter tuning.**
>
> We will add these important sensitive analysis to camera ready version.
>
> ---
>
> ## **2. Explanation of Cold Stop Mechanism**
>
> ### **2.1 How Cold Stop Works**
> Cold Stop does not fundamentally alter the model’s reasoning ability or enable it to produce correct reasoning paths where it previously could not. Instead, its purposes are:
> 1. **Make the answer easy to parse**: By stopping reasoning once the model reaches high confidence, Cold Stop ensures that correct answers are output cleanly and easily to parse by our evaluation scripts automatically.
> 2. **Reducing Token Usage**: It halts unnecessary reasoning steps once the model has already reached the correct conclusion, thereby saving tokens.
>
> **Therefore, for those samples where the judgment changed from incorrect to correct after using Cold Stop, the correct answer has already been output before stopping.**
>
> In addition, since generation rarely happens in baseline (CoT), Cold Stop will not improve the accuracy.
>
> Below is a simple example to illustrate how Cold Stop works. We only show the top 1 token of Soft Thinking for visualization:
> ```
> Question:
> 1+1=?
>
> Without cold stop:
> <think>Let me solve this question. … (some thinking) …. Therefore the correct answer is 2. Emmm, let me verify my answer in another way. Let’s Let’s Let’s Let’s Let’s …  (repetition occurs until the maximum number of tokens is reached)
>
> With cold stop:
> <think>Let me solve this question. … (some thinking) …. Therefore the correct answer is 2. Emmm, let me verify my answer in another way. Let’s Let’s Let’s (cold stop here)</think> The final answer is 2.
> ```
> For the example without cold stop, the correct answer appears in the thinking process but not in the output, and can not be parsed for evaluation. With Cold Stop, we stop the generation collapse properly and let the model conclude so that we can parse the answer.
>
> ### **2.2 Improvement from Cold Stop**
> Accuracy improvements stem from Soft Thinking itself, which enables the model to explore abstract and parallel reasoning paths to find correct solutions. Cold Stop primarily serves as a utility to enhance token efficiency and prevent collapse during inference. Because it cannot make the model generate more content, it will instead reduce the generated content.
>
> Regarding token efficiency, we analyzed the token reductions contributed by Soft Thinking and Cold Stop. The following table summarizes the results:
>
> | Method                      | Accuracy (AIME2024) | Length All | Length Correct | Accuracy (LiveCodeBench) | Length All | Length Correct |
> | --------------------------- | ------------------- | ---------- | -------------- | ------------------------ | ---------- | -------------- |
> | CoT (baseline)              | 76.88               | 13449      | 12080          | 62                       | 13743      | 9986           |
> | Soft Thinking w/o Cold Stop | 73.33               | 12991      | 9457           | 56.98                    | 13360      | 6877           |
> | Soft Thinking w/ Cold Stop  | 83.33               | 11445      | 10627          | 62.72                    | 12537      | 7535           |
>
> Both Soft Thinking (with or without Cold Stop) reduce token usage in terms of total and correct solution length. Cold Stop further mitigates generation collapse, ensuring correct answers are cleanly output while reducing tokens. **As a result, accuracy improvement is from Soft Thinking and efficiency improvement is from both Soft Thinking and Cold Stop.**
>
> For baseline (CoT), generation collapse (repetition) is rare. Since Cold Stop is a method to let model generate output faster, it will not improve the accuracy.
>
> ---
>
> ### **3. Missing Related Work**
> We appreciate the reviewer pointing out missing references. In the camera-ready version, we will update the "Related Work" section to include relevant papers such as:
>
> - Implicit Chain of Thought Reasoning via Knowledge Distillation
> - From Explicit CoT to Implicit CoT: Learning to Internalize CoT Step by Step
> - Follow-up works building on these ideas
>
> Thank you for highlighting this oversight, and we will ensure these contributions are properly acknowledged.
>
> ---
>
> ### **4. Paper Formatting**
> Regarding the term "Visualization of Shorted Example," "Shortened" is a more precise term to describe the concise reasoning chains generated by Soft Thinking. We will update this phrasing in the final version.
>
> ---
>
> ### **Acknowledgment**
> We sincerely respect and appreciate the hard work and professionalism demonstrated in your review. Your thoughtful feedback has been instrumental in improving our paper, and we hope the explanations provided above address all concerns. We look forward to your feedback on our revisions and thank you again for your valuable time and effort.

---

> > ### Author Response · Authors · 2025-08-08
> >
> > Dear Reviewer,
> >
> > Thank you again for your thoughtful comments on our manuscript. We understand you have a very busy schedule and truly appreciate any time you can spare. When convenient, could you kindly review our responses and let us know whether they sufficiently address your concerns? We appreciate your time and guidance.

---

> > > ### Author Response · Authors · 2025-08-09
> > >
> > > Dear Reviewer,
> > >
> > > This is a gentle reminder that only a few hours remain in the author–reviewer discussion period. Our exchange has not yet concluded, and we are still awaiting your response. If you need any further clarification, please let us know; otherwise, please let us know whether your questions have been fully addressed.
> > >
> > > Thank you very much for your insightful guidance and the generous time you have devoted to our work.
> > >
> > > With highest respect!!!

---

> ### Comment · Area_Chair_EYC9 · 2025-08-04
> **Does the response address your concerns?**
>
> Hi, when you have a moment, could you please take a look at the authors' response to see whether it addresses your concerns, or if there are still points that remain unclear? Thank you.

---

> > ### Author Response · Authors · 2025-08-06
> >
> > Dear Reviewer zaBE,
> >
> > Thank you once again for your valuable comments on our submission. As the discussion phase is approaching its end, we would like to kindly confirm whether we have sufficiently addressed your concerns. If there are any remaining questions or areas requiring further clarification, please do not hesitate to let us know. Your suggestions have been invaluable in improving our paper, and we would greatly appreciate your reply.
> >
> > Thank you for your time and constructive feedback.
> >
> > Sincerely,
> >
> > Authors

---

### Comment · Area_Chair_EYC9 · 2025-08-02
**The discussion phase has begun!**

Dear reviewers,

The rebuttal phase has now concluded. I encourage you to take a moment to carefully read the authors' responses and consider whether they address your concerns. Your insights during the discussion phase are important for reaching a fair and informed decision. If your opinion has changed based on the response or your reviews, please share your updated thoughts.

Thank you again for your time and contributions!

Best regards, Your Area Chair EYC9.

---

### Note · Authors · 2025-08-14

We sincerely appreciate the efforts of the AC and reviewers during the rebuttal period. In response to the main concerns raised by the reviewers, we have provided the following clarifications and additional analyses:

- **Hyperparameter sensitivity concerns**: Supplementary experiments demonstrate strong robustness to hyperparameters, showing consistent performance across different parameter settings. (zaBE, 81RM, txtk)

- **Cold stop mechanism clarification**: We further clarified the cold stop mechanism, which saves tokens and helps evaluation scripts better extract results, contributing to both efficiency and reliability. (zaBE, 81RM, txtk)

- **Parallel Thinking explanation**: Detailed clarifications were provided on how our method enables parallel thinking, supported by theoretical analysis, empirical results, and case studies from our paper. (m4ot)

- **OOD generalization**: We explained that our method generalizes well to OOD problems, where "OOD" refers to concept tokens rather than entirely new domains, demonstrating the flexibility of continuous concept space reasoning. (81RM, txtk)

---

We also want to highlight the contributions of this work as follows:

- We introduce **Soft Thinking**, a training-free method that enables LLMs to reason in continuous concept space using probability-weighted token embeddings, breaking the constraints of discrete token-based reasoning.

- Empirical and theoretical analyses demonstrate how continuous concept tokens implicitly explore multiple reasoning paths, leading to improved accuracy and efficiency compared to standard Chain-of-Thought methods.

- Comprehensive experiments across diverse mathematical and coding benchmarks validate the effectiveness of Soft Thinking, achieving up to 2.48 points improvement in pass@1 accuracy while reducing token usage by up to 22.4%.

Again, we deeply appreciate the hard work of the AC and reviewers and generous service.

---

### Decision · Program_Chairs · 2025-09-17

**Decision:**

Accept (poster)

**Comment:**

This paper introduces Soft Thinking, a training-free reasoning framework for large language models that shifts reasoning from the discrete token space (as in traditional Chain-of-Thought) to a continuous concept space. Instead of sampling a single token and embedding it, the method constructs concept tokens—probability-weighted mixtures of embeddings drawn from the output distribution (with top-k and top-p constraints). This enables the model to implicitly explore multiple reasoning paths in parallel, improving efficiency and robustness. After rebuttal and discussion stages, all reviewers gives positive comments. Thus I also recommend to accept this paper.